# Isolation, Identification and Pollution Prevention of Bacteria and Fungi during the Tissue Culture of Dwarf Hygro (*Hygrophila polysperma*) Explants

**DOI:** 10.3390/microorganisms10122476

**Published:** 2022-12-15

**Authors:** Weijie Li, Guanglong Cao, Mengqian Zhu, Yilin Zhang, Rong Zhou, Zhenyang Zhao, Yaning Guo, Wanli Yang, Bo Zheng, Jiabo Tan, Yanling Sun

**Affiliations:** Marine Science and Engineering College, Qingdao Agricultural University, Qingdao 266237, China

**Keywords:** *Hygrophila polysperma*, tissue culture, bacteria, fungi, contamination

## Abstract

Microbial contamination causes serious damage in plant tissue culture, and attention is always being paid regarding how to control and prevent the unwanted pollution. Dwarf hygro (*Hygrophila polysperma*) is a popular ornamental aquatic plant and its tissue culture has been reported, but the microbial pollution and the cure of microbial pollution was unknown. In this study, a number of bacteria and fungi were isolated from contaminants in MS culture media. Based on the 16S rDNA and ITS sequencing, it was identified that fifteen bacteria belong to *Bacillus*, *Enterobacter, Pantoea, Kosakonia, Ensifer* and *Klebsiella*, and three fungi belong to *Plectosphaerella, Cladosporium* and *Peniophora*, respectively. In addition, some drugs were further tested to be free of the bacteria and fungi pollution. The results revealed that 10 μg/mL of kanamycin, 5 μg/mL of chloramphenicol, and 0.015625% potassium sorbate could be applied jointly in MS media to prevent the microbial pollution, and the survival rate of *H*. *polysperma* explants was highly improved. This study reveals the bacteria and fungi species from the culture pollution of *H*. *polysperma* and provides a practical reference for optimizing the tissue culture media for other aquatic plants.

## 1. Introduction

Plenty of aquatic species are naturally propagated at a long cultivation period, or subjected to unstable growth because of water transmitted diseases and variable environmental conditions [1]. However, planting the tissue culture of aquatic plants could break these limitations. Under artificial control, the explants from leaves and shoots, embryos or protoplasts were cultured on sterile media in vitro, and further, newly differentiated into a whole plant. In tissue culture, explants could be reproduced with high speed and heredity stability [2,3,4]. The success of in vitro plant culture was contributed to many biotic or abiotic factors, including the explants’ physiological status, the favorable culture media and ecological factors, etc. [5]. Among them, the control of microbial contamination was crucial. When the frequent pollution occurred in culture media, the explants could become withered and even die, and finally, it failed to be in progress of the tissue culture. Multiple factors caused the microbial pollution of plant tissue culture. In vitro, the incomplete sterilization, culture rooms and transfer fields are all possible contaminated sources [6]. In vivo, plants naturally keep a symbiotic relationship with diverse microorganisms, described as endophytes, including bacteria, fungi, and archaea [7,8,9]. Increasingly, a variety of endophytes were reported on high land plants, including *Bacillus*, *Pseudomonas, Enterobacter*, *Rhizobium* and *Klebsiella*, *Agrobacterium*, *Methylobacterium* spp., etc. [10,11,12,13,14]. Few were reported in aquatic plants, and it was yet unknown about the endophytes. Early in 1976, some mesophilic bacteria were identified from the purchased ornamental aquarium plants from retail outlets, and the dominant bacteria was *Pseudomonas aeruginosa* (*P*. *aeruginosa*) [15]. Recently, various drugs had been tested to avoid the pollution, including antibiotics, chemical compounds, Chinese traditional medicines, and nano materials [3,5,6,16]. However, these studies were mainly focused on high terrestrial plants [17,18,19]. There are few reports on the pollution and prevention of tissue culture in aquatic plants.

Aquatic plants are broadly defined as plants that undergo chloroxygenic photosynthesis, containing numerous single-cell and multicellular species with abundant populations [20]. They play ecological and functional roles in various aspects, such as assimilating excessive nutrients (e.g., nitrogen and phosphorus) in eutrophic waters [21,22], removing heavy metals and nanoparticles in wastewater [23,24,25,26,27], and maintaining mutual benefit with aquatic animals [28], as well as future biofuels [29] and medicinal plants [30,31,32,33]. Importantly, most of them were beautiful ornamental plants in the aquarium, with high commercial value [2,3,4,34,35]. *Hygrophila polysperma* (or *Hygrophila rubella*), known as dwarf hygro, belongs to Acanthaceae, the *Hygrophila* genus. *H. polysperma* has lanceolar leaves, sawtooth edge and clear veins, and leaves of plants often maintain a variable color; red at the top and green at the middle and base, depending on high light intensity and abundant nutrients. As an emergent macrophyte, it was usually chosen as a middle or background grass in aquariums. It is not only a popular aquarium plant, but also a medicinal plant and bioindicator for the control of algae and toxicities, it being in increasing market demand [3,31]. The tissue culture technique is an indispensible approach for the rapid propagation of *H*. *polysperma*. However, the microbial contamination frequently emerged in tissue culture of *H*. *polysperma*, and caused high mortality of explants. As previously reported, adventitious shoots of *H*. *polysperma* were regenerated in Murashige and Skoog (MS) culture media, and Amoklavin was applied for preventing the bacterial pollution [3]. Yet, it could not inhibit the fungi growth and was not convenient to acquire from the local farm in China. Thus, this study is aimed to reveal the bacteria and fungi species from the contaminants in culture media and further to seek the drugs to control them effectively for the success of tissue culture in *H*. *polysperma*.

## 2. Materials and Methods

### 2.1. Plant Material and Culture Conditions

*H*. *polysperma* were purchased from a local aquatic plant farm in Shandong province, China. Five-six cm long twigs of *H*. *polysperma* containing 5–6 nodes from plants with attached leaves were firstly washed for 5 min under tap water. In the clean bench, the leaves were removed from the twigs and the twigs were cut into the 4–5 cm long segments, containing 2–3 nodes, then it was surface-sterilized with 75% alcohol for 30 s, and rapidly transferred into 2% NaClO solution for 10 min, followed by rinsing three times with sterilized distilled water via continuous stirring. Subsequently, the sterilized twigs were cut into 0.5–1 cm explants, and cultured in 240 mL culture vessels (6 × 9 cm) containing MS medium for several weeks [36], which was supplemented with 3% sucrose, 1.0 mg/L of 6-KT and 1.0 mg/L of IBA, and solidified with 0.65% agar with the pH adjusted to 5.8 ± 0.1 before autoclaving (118 kPa atmospheric pressure, 121 °C for 20 min). Each culture vessel contained two-four shoot explants. All cultures were incubated at 26 ± 1 °C and 60–70% humidity under a 16 h light photoperiod (5000 lux) using white fluorescent lights in the climatic chamber (Ningbo, China).

### 2.2. Isolation and Identification of Bacteria

Based on the phenotypical observation, the bacterial contaminants, frequently appeared in MS media within a month. To culture the appearing bacteria as much as possible, bacteria were streaked with inoculation loops on the common microbial media, Luria-Bertani (LB) and tryticase soy broth (TSB) agar plates to incubate at 26 °C for 48 h, respectively. Then, single bacterial colonies were selected and restreaked on their preferential media again. When the morphology of all colonies on agar plates was identical, and the pure culture of bacterial isolate was obtained, all isolates were maintained separately in LB or TSB broth with 15% glycerol at −80 °C.

### 2.3. Isolation and Identification of Fungi

Several fungi also appeared in MS media. Based on the phenotypical observation, the typical fungi were chosen and inoculated on potato dextrose agar (PDA) plates, and then incubated at 26 °C for 72 h. After several inoculations, the fungal pure culture was obtained and stored on PDA plates at 4 °C until use.

The fungal shapes were detected by the scotch cellophane tape method using a lacto-phenol cotton blue (LPCB) solution to stain the fungi blue. One cellophane tape containing fungal sample was pressed against the surface of the glass slide, while another cellophane tape was pasted at its non-ground glass end. Then a drop of LPCB was placed on the side where the tape crosses. After several minutes, LPCB was permeated into the fungal bodies, and the fungal morphology was observed with a microscope and pictures were taken.

### 2.4. Molecular Analysis of Bacteria and Phylogenetic Tree Construction

The bacterial cells were washed twice with sterile phosphate buffer (pH 7.4), and genomic DNA were extracted using a FastPure^®^ bacteria DNA isolation mini kit (Vazyme, Nanjing, China) according to the manufacturer’s protocol and stored at −20 °C until use, which was used as a template for PCR reaction. The PCR reaction contained 10.0 μL of 2 × Taq PCR Mastermix, 0.4 μL each of 10 μM sense primer (27F) and anti-sense primer (1492R), 1.0 μL of 500 ng of template DNA, in a final volume made up to 20 μL with sterile distilled water. PCR was performed for all isolates to amplify the 16S rDNA gene using the universal primers 27F (Sense primer, 5′-AGAGTTGATCATGGCTCA-3′) and 1492R (Anti-sense primer, 5′-GGTTCACTTGTTACGACTT-3′), as described previously [37]. The amplifications were carried out in a thermal cycler (BioRad, USA), with a pre-cycle denaturation at 95 °C for 5 min followed by 34 serial cycles of 94 °C for 30 s, 55 °C for 30 s, and 72 °C for 1 min, with a final extension at 72 °C for 7 min. The PCR products were analyzed by 1.2% (*w*/*v*) agarose gel (containing GelRed) electrophoresis in 1% tris–acetic acid–EDTA buffer. Gels were visualized and photographed under UV illumination. The PCR products were sent to the Ruibo Xingke biotechnology Co., Ltd. (Qingdao, China) and conducted by Sanger sequencing. Then, the 16S rDNA sequences were submitted to the NCBI GenBank database and blasted using the BLASTn algorithm. The identification of a specie of bacteria was more than 97% nucleotide identity for the 16S rDNA sequence.

A phylogenetic tree was created using the multi-sequence alignment generated via the CLUSTAL X program in Mega 6.0 by the neighbour-joining method [38]. The genetic distances matrix was obtained using Kimura’s two-parameter model [39], and bootstrap values from 1000 replicates are displayed as percentages.

### 2.5. Molecular Analysis of Fungi

The fungi hyphae were collected and grounded with sterile glass bead, and genomic DNA was extracted using a fungal DNA isolation Kit (Solarbio, Beijing, China). DNA as template was used for the PCR reaction to amplify the internal transcribed spacer (ITS) sequence of fungal ribosomal genes [40], using the primers of ITS-F (Sense primer, 5′-TCCGTAGGTGAACCTGCGG-3′) and ITS-R (Anti-sense primer, 5′-TCCTCCGCTTATTGATATGC-3′), followed by the similar protocol described as above in Section 2.4.

### 2.6. Screening Test for Bacterial Inhibition

Bacteria were grown at 26 °C in LB broth for 24 h. Bacterial growth was assessed for two weeks on LB broth and agar supplemented with 50 μg/mL of kanamycin (Kan), 100 μg/mL of ampicillin (Amp), 250 μg/mL of erythromycin (Ery), 200 μg/mL of spectinomycin (Spc), or 25 μg/mL of chloramphenicol (Chl) at 26 °C, respectively. The antibiotic tolerance levels were examined and recorded as sensitive (−) or resistant (+) accordingly.

### 2.7. Screening Test for Fungal Inhibition

The fungi growth was assessed for two weeks on PDA agar plates supplemented with 0.015625% (*w*/*v*), 0.03125% (*w*/*v*), 0.0625% (*w*/*v*), 0.1% (*w*/*v*), 0.125% (*w*/*v*), 0.25% (*w*/*v*), 0.5% (*w*/*v*) and 1% (*w*/*v*) potassium sorbate (PS) at 26 °C, respectively. Equally, a series concentration of sodium benzoate (SB) or sodium diacetate (SD) solution, as well as nanoscale silver (nano-Ag) was tested for the inhibition of fungi, respectively.

### 2.8. The Influences of the Antimicrobial Agents on Explants

The effects of antimicrobial agents on explants were tested. On the one hand, 50 μg/mL of Kan and 25 μg/mL of Chl (Kan + Chl), 1/2 (Kan + Chl), 1/5 (Kan + Chl) and 1/10 (Kan + Chl) were added into MS media, then the inhibition of bacteria was examined and the explants’ growth was observed for two weeks. On the other hand, a series of concentration (0.015625%, 0.03125%, 0.0625%, 0.1%, 0.2% and 0.5%) of PS, SB and SD were investigated on the inhibition of fungi and influences on the explants’ growth for one month. Subsequently, the appropriate concentration of antimicrobial agents was determined. Then, the explants or seedlings were grown in MS media supplemented with the above appropriate concentration of drugs. Finally, the seedling was transplanted into the soil of the aquarium, and its growth was recorded and photos were taken for two months.

## 3. Results

### 3.1. Phenotypic Profiles of the Microbial Contaminants

Currently, via the tissue culture technique, *H*. *polysperma* plantlets could be rapidly propagated using the young leaf and stem explants in agar-gelled MS media. Although *H. polysperma* explants were sterilized with 75% alcohol and 2% NaClO solution, the microbial contamination still commonly emerged in MS media during the tissue culture. Obviously, the bacteria appear as the milky white, light or yellow clouds (Figure 1A,E,F), droplet and mucoid or dry white sediment (Figure 1C,D,G). Some appeared as a wrinkled broccoli-like pattern, (Figure 1B,G), even in pink or fermentative foam (Data not shown), along with disgusting odors. The fungi were observed with white or green radial villiform, due to their abundant fungal hyphae (Figure 1A,G,H). In addition, once the bacteria and fungi appeared, and it could proliferate rapidly to cover the whole media within three to five days, leading to severe damage for explants. As shown in Figure 1, the explants gradually turned brown and wrinkled, and eventually died.

### 3.2. Isolation and Identification of Bacteria and Fungi

Bacteria were isolated from the contaminants and their pure cultures were obtained, respectively (Figure 2). Used the genomic DNAs as the templates, 16S rDNA genes were amplified by the universal primers, and an approximate 1600 bp long electrophoretic bands were produced (Figure 3A). Then, the 16S rDNA sequences were subjected to NCBI Blast analysis for the species identification. Consequently, these bacteria belong to 15 species and 6 genera, including *Pantoea*, *Kosakonia*, *Bacillus*, *Ensifer*, *Enterobacter* and *Klebsiella*, respectively (Table 1).

Meanwhile, the pure cultures of three fungi were obtained, and genomic DNAs were extracted to be used as a template. The ITS sequences of them were amplified and clear 400 bp long PCR fragments were produced (Figure 3B and Figure 4). Similarly, based on the ITS sequencing analysis, three fungi belong to *Plectosphaerella*, *Cladosporium* and *Peniophora* (Table 2). The fungi *Peniophora* sp. strain and *P*. *oligotrophica* were in hairy and spherical shapes, respectively. Both *Peniophora* sp. and *P*. *oligotrophica* presented a white color, while *Cladosporium crousii* exhibited a green color (Figure 4A–C). Using the scotch cellophane tape method, the morphology was observed in three fungi. The long fungal hyphae of them were stained blue with LPCB solution (Figure 4D–F), and *C*. *crousii* possessed many scattered spores with blue staining (Figure 4F).

### 3.3. Molecular Analysis of Bacteria and Fungi

In order to clarify the evolutionary relationship among the bacteria, the phylogenetic tree was constructed based on the 16S rDNA sequences. The results showed that *Bacillus siamensis*, *B. koreensis* and *B. aryabhattai* B8 formed a single cluster with a close cluster of *Enterobacter* E24. *B*. *amyloliquefaciens* Y5 and *Enterobacter* Glu2 formed a single cluster, while *B. amyloliquefaciens* BV2007 possessed high similarity to *K*. *michiganensis* (Figure 5A).

Moreover, the phylogenic tree in fungi showed that the *Peniophora* sp. strain formed a single cluster with a reference strain *Peniophora* M161, while *P*. *oligotrophica* and *C*. *crousii* formed a single cluster, indicating a close relationship between them (Figure 5B).

### 3.4. Drugs Screening from Antimicrobial Susceptibility Results

Based on the susceptibility tests, 15 isolates were all resistant to 100 μg/mL of Amp and 0.0625% PS, 13 isolates were resistant to 200 μg/mL of Spc, and 7 isolates were resistant to 250 μg/mL of Ery (Table 3; Appendix A). Importantly, the bacteria were all sensitive to 50 μg/mL of Kan, except for *K. michiganensis* and *Enterobacter* Glu2, but they were both sensitive to 25 μg/mL of Chl (Table 3; Appendix A). Thus, a combination of 50 μg/mL of Kan and 25 μg/mL of Chl were selected for potential therapy of bacterial pollution in MS media.

In addition, some drugs, including nano-Ag, PS, SB and SD, were screened for the resistance of fungi. The results showed that PS could completely inhibit the three fungi growths at series concentrations of 0.0125% (*w*/*v*), 0.25% (*w*/*v*), 0.5% (*w*/*v*) and 1.0% (*w*/*v*).

In contrast, less than 0.25% SB was not free of *C*. *crousii* and 0.125% SD could no longer inhibit the three fungi production (Appendix A). No inhibitory effects were observed in nano-Ag, Spc and Chl (Figure 6). Thus, PS had remarkably inhibitory effects on the fungi and was selected for the prevention of fungal pollution in MS media.

### 3.5. The Application of Antimicrobial Agents in Tissue Culture

To avoid the influences on the *H*. *polysperma* explants or seedlings, different concentrations of the above antimicrobial agents were tested. The results showed that a combination of 10 μg/mL of Kan and 5 μg/mL of Chl had good antibacterial activity in MS media and also no side effects on the explants’ growth, while when using lower than those concentrations, severe pollution occurred in MS media (Figure 7A–D; Appendix A).

Moreover, 0.015625% PS had good performance for antifungal activity. In contrast, higher than 0.015625% PS, from 0.03125% to 0.5%, caused worse influences on the seedlings’ growth. Similar effects were observed under the treatments of SB and SD, respectively (Appendix A; Appendix A).

Hence, a combination of 10 μg/mL of Kan and 5 μg/mL of Chl, and 0.015625% PS were well applied in MS media to prevent the microbial pollution. Subsequently, *H*. *polysperma* explants were in normal progress of tissue culture, and finally the *H*. *polysperma* seedlings were transplanted and grew well in the aquarium (Figure 7E,F).

## 4. Discussion

Aquatic plants grew in freshwater and saltwater with varied life cycles and architectures, and their total diversity is abundant [41]. Aquatic macrophytes, belonging to multicellular species, mostly grew in freshwater. The macrophytes commonly have soft stems with branch shapes of low and creeping clumps, and exhibit in free-floating, emergent or submerged forms, such as *Pistia stratiotes*, *Myriophyllum aquaticum*, *Egeria densa* and *Typha angustifolia*. Aquatic plants could usually reproduce rapidly via vegetative means. Among them, the tissue culture was used as an important method for mass production in business. Plenty of aquatic plants had been reported, including free-floating *Nymphoides indica* [42] and *P*. *stratiotes* [24], rooted *Anubias barteri* [43], emergent *Nymphaea lotus* [1], submerged *Rotala rotundifolia* [4], *Ludwigia repens* [44], *Cryptocoryne lucens* [45,46] and *C*. *wendtii* [47]. In *H*. *polysperma*, it was previously found that multiple adventitious shoots were easily induced from the leaf explants, with the additive of 0.1 mg/L TDZ and 0.10 mg/L IBA in MS media [3]. In this study, 1.0 mg/L 6-KT and 1.0 mg/L IBA were added in MS media, and adventitious shoots were well induced from the stem explants, hinting that callus inducing from different explants was dependent on the different concentration of phytohormones.

Plants closely associated with “endophyte” living within plant tissues that are not eradicated after surface sterilization, can interfere with or inhibit the initiation and growth of in vitro cultures [48]. The endophytes usually keep symbiotic relationships with plants for mutual benefits. Accidentally, over-reproduction of endophytes caused the bacterial pollution. The endophytes from a variety of plants had been previously observed by microscopy technique via fluorescent in situ hybridization (FISH) [49,50,51,52,53] and identified by 16S rDNA sequencing [37,50,54,55]. For instance, *Bacillus*, *Pseudomonas, Fusarium, Burkholderia, Rhizobium* and *Klebsiella* were identified as the most dominant genera in crops, including wheat, maize, rice and bean, etc. [10]. *Pantoea*, *Enterobacter, Agrobacterium*, and *Methylobacterium* spp., were identified as endophytes in grape, banana, cotton, clover and potato [11,12,13,14]. Significantly, *Pseudomonas* and *Burkholderia* were found to fight against pathogens and pests as antagonists, and *Klebsiella* participated in nitrogen fixation [8,14,51,56]. Besides, *Pseudomonas* and *Bacillus* were found colonized on roots of plants [49], and probably migrated upward and reached different tissues of plants later [10,57]. Especially, *Proteobacteria*, including *Pseudomonas* and *Enterobacter* was predominantly isolated from the callus stocks of grapevine, barley, tobacco and Arabidopsis [9]. In the ornamental aquarium plants, *Bacillus*, *Enterobacter*, *Klebsiella* and *Pseudomonas* were detected early [15]. Consistently with the previous reports, *Bacillus* spp., *Enterobacter, Pantoea* and *Klebsiella* strains in the present study were identified from the contaminants in the MS media of *H. polysperma*, suggesting a common bacterial pollution among different aquatic and land plant species. Especially, *Kosakonia* and *Ensifer* strains were firstly reported in the tissue culture of *H. polysperma* in this study. Whether these bacteria are endophytes or not needs to be further investigated via whole-metagenome profiling (WMG), which could clarify the taxonomic and functional diversity of bacteria [8,10,50,54]. Moreover, three kinds of fungi belong to *Plectosphaerella*, *Cladosporium* and *Peniophora*, not previously reported in eukaryotic endophytes in plants [54], and so it was inferred that they were derived from the external working environment.

Excessive proliferation of endophytes and the microbial pollution from the external conditions would severely hinder the progress of tissue culture. Traditionally, plant material was surface-disinfected with a treatment of 70–75% ethanol, 2% NaClO, or 0.1% HgCl_2_. However, these disinfectants are also harmful to plant cells and needed to be used at an appropriate dose and time. In this case, certain types of microorganisms, such as *Pseudomonas*, *Agrobacterium* and *Klebsiella* spp. [58], are likely not to be destroyed; the endophytes might be also protected from the surface disinfection treatment. Accordingly, in this study, the *H. polysperma* explants were heavily infected with multiple bacteria and fungi, which occurred in different colors and shapes. In order to eliminate the unwanted microbes, several drugs were tested. As previously reported, antibiotics induced side effects on the chrysanthemum and tobacco growth in vitro [16], so the appropriate inhibitory concentrations were determined to avoid the negative influences on the explants [1]. Expectedly, a mixture of 10 μg/mL of Kan, 5 μg/mL of Chl and 0.015625% PS could inhibit the microbial infection and was well applied in the MS media of *H. polysperma*. Usually, the explants are not always infected at the establishment stage, but frequently this occurs later in the multiplication stage after several subcultures [1]. Thus, the medicine might be more needed at the later propagation than at the early initial stage. Recently, wide ranges of nanoparticles (NPs) were reported to possess antimicrobial activity, including NPs of silver (Ag), silicon dioxide (SiO_2_), titanium dioxide (TiO_2_), and Zinc oxide (ZnO) [1]. However, in this study, no inhibitory effect was found in nano-Ag against the fungi, possibly due to its low additive dose.

Noteworthily, a number of endophytic bacteria were favorable for promoting plant growth, associated with an increase in nutrient availability and the production of the endogenous phytohormone [49]. For instance, auxin–producing bacterial strains from *Pseudomonas*, *Enterobacter*, *Rhizobium*, *Bradyrhizobium*, *Bacillus*, *Methylobacterium*, *Rhodococcus*, *Acinetobacter* and *Microbacterium* had been reported to improve the endogenous IAA level of the *Triticum aestivum* plant [59,60]. Considering the positive effect of endophytic bacteria in plants, it might be rational to keep a balance of the total microbial community rather than completely kill them during the tissue culture [49]. In future, it might be helpful to solve this problem from the next-generation sequencing (NGS)-mediated explorations, which could uncover the microbial communities profile and provide considerable taxonomic diversity with functional implications.

In conclusion, this report revealed that the bacteria, commonly in *Bacillus*, *Enterobacter, Pantoea* and *Klebsiella*, and the fungi, containing *Peniophora* sp., *P*. *oligotrophica* and *C*. *crousii*, were reported in the contaminated MS media of *H. polysperma* explants. Three antimicrobial agents, 10 μg/mL of Kan, 5 μg/mL of Chl and 0.015625% PS, were screened out to be applied jointly in culture media to prevent the microbial pollution, which provided a practical reference for the modified MS media on the tissue culture of other aquatic plants.

## Figures and Tables

**Figure 1 microorganisms-10-02476-f001:**
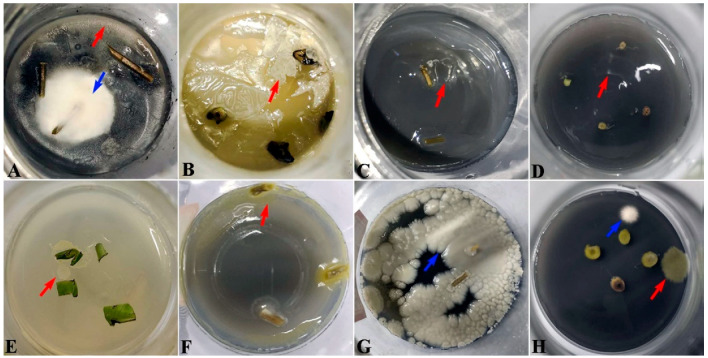
Phenotypes of microbial contamination in tissue culture flasks. (**A**–**F**), Various kinds of bacteria were grown around the explants in MS contaminated media during tissue culture. (**A**,**G**,**H**), The fungi of mixed species appeared in MS contaminated media during tissue culture. The red arrows indicate the bacteria and the blue arrows indicate the fungi.

**Figure 2 microorganisms-10-02476-f002:**
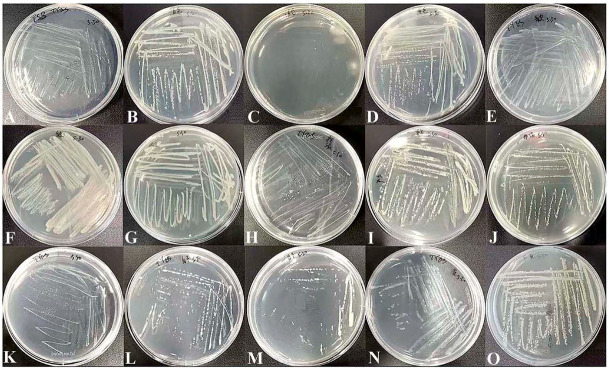
Morphological characterizations and the 16S rDNA amplification of bacteria obtained from the contaminated media. Morphology of 15 bacterial isolates grown in LB or TSB plates, respectively. (**A**) *Kosakonia oryzendophytica* strain (*Ko*). (**B**) *Pantoea* sp. strain BHUJPCS-26 (BH). (**C**) *Bacillus amyloliquefaciens* strain BV2007 (BV). (**D**) *Bacillus siamensis* strain (*Bsi*). (**E**) *Bacillus subtilis* strain (*Bsu*). (**F**) *Bacillus amyloliquefaciens* strain Y5 (Y5). (**G**) *Bacillus koreensis* strain (*Bk*). (**H**) *Bacillus aryabhattai* strain B8W22T.35 (B8). (**I**) *Ensifer* sp. strain BO-30 (BO). (**J**) *Bacillus zanthoxyli* strain (*Bz*). (**K**) *Bacillus aryabhattai* strain WH6 (*Ba*). (**L**) *Enterobacter*
*cloacae* strain PR3 (*Ec*). (**M**) *Klebsiella michiganensis* strain (*Km*). (**N**) *Enterobacter* sp. strain Glu2 (Glu2). (**O**) *Enterobacter* sp. strain E24 (E24).

**Figure 3 microorganisms-10-02476-f003:**
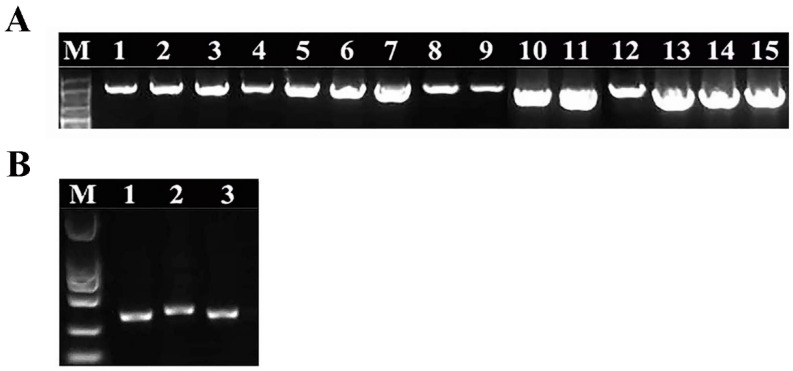
16S rDNA amplification of the bacteria and ITS amplification of fungi isolated from unwanted contaminants in culture media. (**A**) Lanes 1–15 indicated *Ko*, BH, BV, *Bsi*, *Bsu*, Y5, *Bk*, B8, BO, *Bz*, *Ba*, *Ec*, *Km*, Glu2 and E24, and DNA extracted from 15 bacterial isolates, respectively. 16S rDNA was amplified from the DNA template and strong PCR amplification fragments were produced as shown, as approximately 1600 bp long electrophoretic bands. (**B**) Lanes 1–3 indicated *Peniophora* sp. strain (*Peniophora* sp., *Pe*), *Plectosphaerella oligotrophica* strain (*P*. *oligotrophica*, *Po*) and *Cladosporium crousii* strain (*C. crousii*, *Cc*), and DNA extracted from three fungal isolates, respectively. ITS was amplified from the DNA template and a strong PCR amplification fragments were produced as shown approximately 400 bp long electrophoretic bands**.** Lane M, 2000 bp DNA ladder.

**Figure 4 microorganisms-10-02476-f004:**
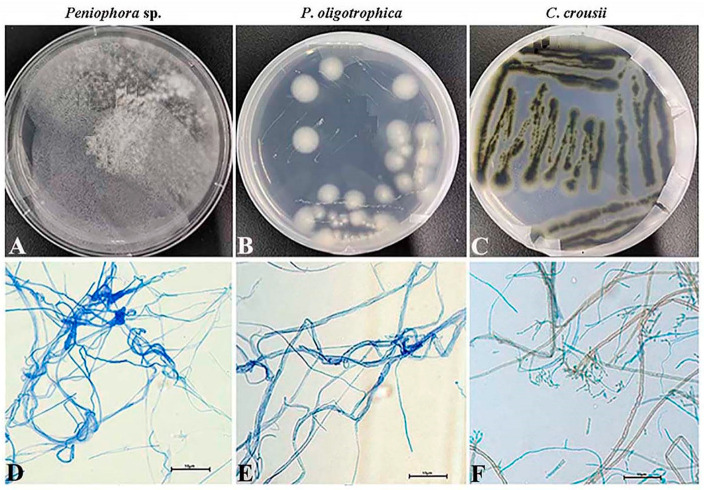
Phenotypic characterizations of fungi obtained from the contaminated media. Morphology of three fungi grown in PDA plates, respectively. (**A**–**C**) indicates the fungus *Pe*, *Po* and *Cc*, respectively. the morphologies of the fungi were observed by scotch cellophane tape method, which presented in (**D**–**F**), respectively.

**Figure 5 microorganisms-10-02476-f005:**
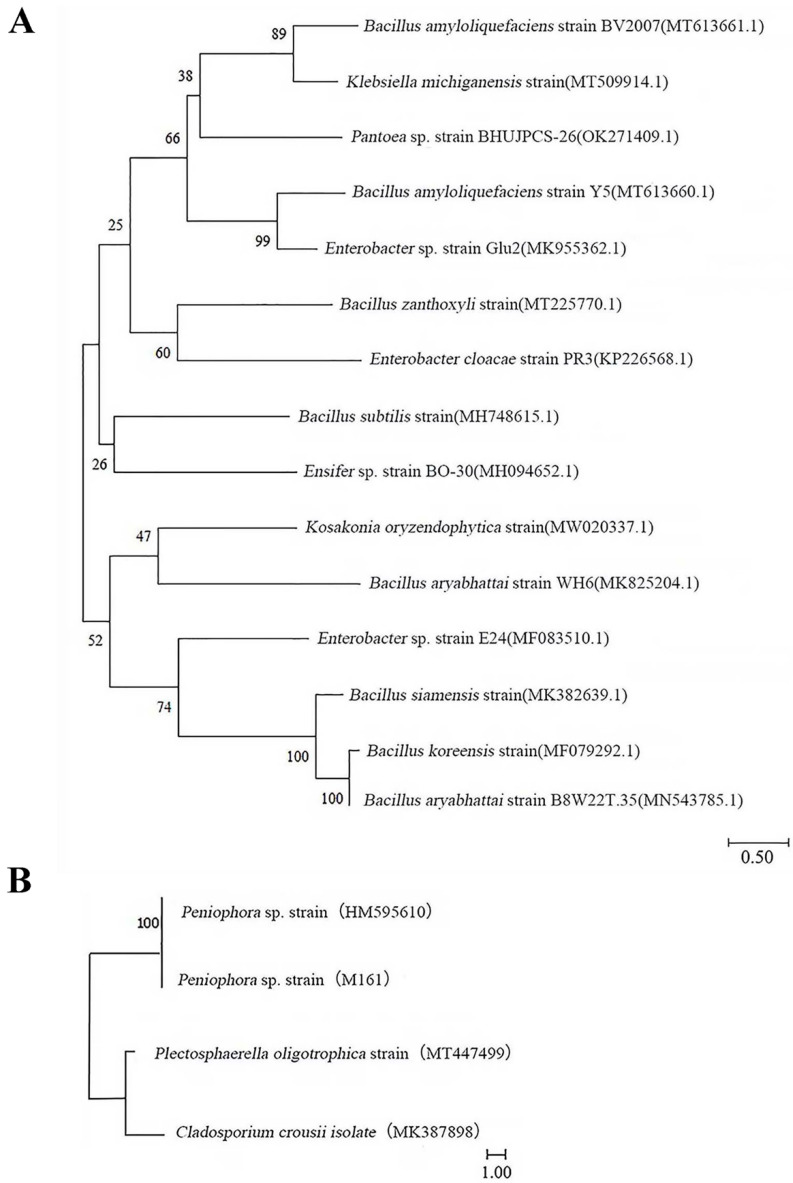
Phylogenetic tree of the detected bacteria and fungi. Phylogenetic trees were constructed and the relationships were showed among the detected bacteria (**A**) and fungi (**B**) obtained from unwanted contaminants in culture media, respectively. The branching pattern was generated by the neighbour-joining method. Bootstrap analysis was performed with 1000 repetitions. Bar, 0.5 or 1.0 nucleotide substitutions per site. GenBank accession numbers are given in parentheses.

**Figure 6 microorganisms-10-02476-f006:**
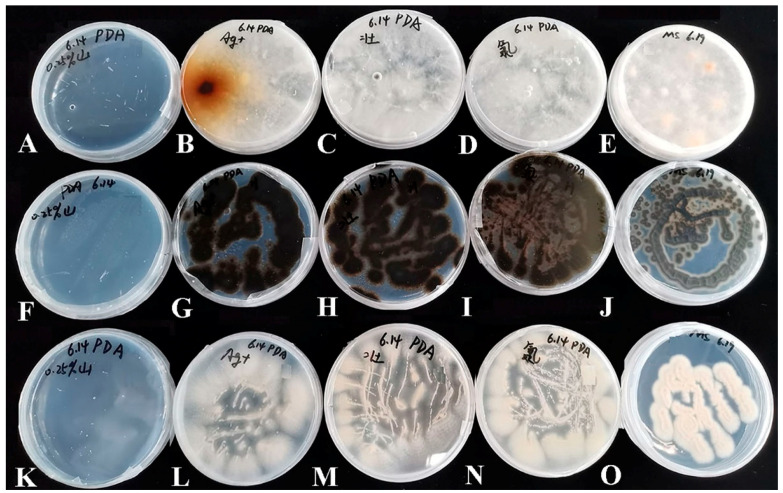
Effects of antimicrobial agents on the three species of fungi. (**A**–**E**) indicated the fungi *Pe* confronting with 0.25% PS, 1000 ppm nanoscale silver (nano-Ag), 200 μg/mL of Spc, 25 μg/mL Chl and the control (MS media), in turn. (**F**–**J**) indicated the fungi *Cc* and (**K**–**O**) indicated the fungi *Po* strain treated with the same antimicrobial agents, respectively.

**Figure 7 microorganisms-10-02476-f007:**
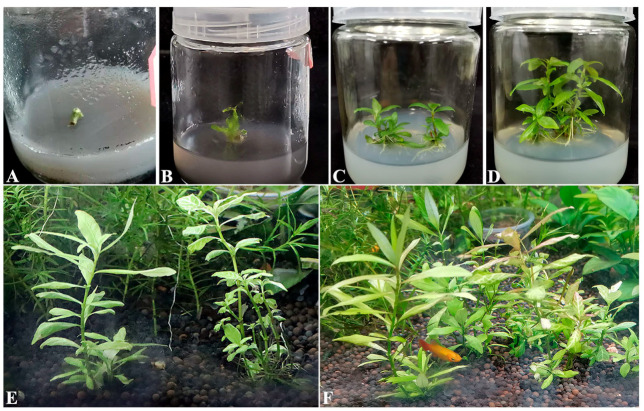
The phenotypic response of *H*. *polysperma* explants to the antimicrobial agents. (**A**) The stem explants of *H*. *polysperma* were grown in the MS medium supplemented with 5 µg/mL of Chl, 10 µg/mL of Kan and 0.015625% PS. Multiple shoots were initiated from the explants after two weeks (**B**), and then the seedlings were grown under the treatment of three drugs for four weeks (**C**) and six weeks (**D**). Subsequently, the seedlings were acclimatized and transported into aquariums for one month (**E**) and two month (**F**), respectively.

**Table 1 microorganisms-10-02476-t001:** The species analysis of bacteria from the contaminated media.

Bacterial Names	Abbreviations	Characteristics
*Kosakonia oryzendophytica* strain	*Ko*	white
*Pantoea* sp. strain BHUJPCS-26	BH	white
*Bacillus amyloliquefaciens* strain BV2007	BV	wrinkled
*Bacillus siamensis* strain	*Bsi*	wrinkled
*Bacillus subtilis* strain	*Bsu*	Mucus
*Bacillus amyloliquefaciens* strain Y5	Y5	Mucus
*Bacillus koreensis* strain	*Bk*	Mucus
*Bacillus aryabhattai* strain B8W22T.35	B8	Mucus
*Ensifer* sp. strain BO-30	BO	droplet
*Bacillus zanthoxyli* strain	*Bz*	Dry white sediment
*Bacillus aryabhattai* strain WH6	*Ba*	Dry white sediment
*Enterobacter**cloacae* strain PR3	*Ec*	Light yellow
*Klebsiella michiganensis* strain	*Km*	Yellow
*Enterobacter* sp. strain Glu2	Glu2	Yellow
*Enterobacter* sp. strain E24	E24	Yellow

**Table 2 microorganisms-10-02476-t002:** The species analysis of fungi from the contaminated media.

Fungal Names	Abbreviations	Characteristics
*Peniophora* sp. strain	*Pe*	White villus
*Plectosphaerella oligotrophica* strain	*Po*	Tasteless white villus
*Cladosporium crousii* strain	*Cc*	Tasteless green villus

**Table 3 microorganisms-10-02476-t003:** Antimicrobial susceptibility testing of the bacterial strains isolated from the contaminated culture media.

Bacterial Name	Kan	Amp	Ery	Spc	Chl	PS
*K*. *oryzendophytica*	−	+	+	+	+	+
*Pantoea* BH	−	+	+	+	+	+
*B*. *amyloliquefaciens* BV	−	+	−	+	+	+
*B*. *siamensis*	−	+	−	+	−	+
*B*. *subtilis*	−	+	−	+	−	+
*B*. *amyloliquefaciens* Y5	−	+	−	−	−	+
*B*. *koreensis*	−	+	−	+	−	+
*B. aryabhattai* B8	−	+	−	−	−	+
*Ensifer* BO	−	+	+	+	−	+
*B. zanthoxyli*	−	−	−	−	−	+
*B. aryabhattai* WH6	−	+	−	+	−	+
*E. cloacae*	−	+	−	+	−	+
*K. michiganensis*	−	+	+	+	+	+
*Enterobacter* Glu2	+	+	+	+	−	+
*Enterobacter* E24	+	+	+	+	−	+

Note: “+” indicates that a bacterial strain could not be inhibited by the antibiotic; “−” means that a bacterial strain could be inhibited by the antibiotic. Kan, 50 μg/mL of kanamycin. Amp, 100 μg/mL of ampicillin. Ery, 250 μg/mL of erythromycin. Spc, 200 μg/mL of spectinomycin. Chl, 25 μg/mL of chloramphenicol. PS, 0.0625% potassium sorbate.

## Data Availability

Not applicable.

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
