# Peer review of "Isolation, Identification and Pollution Prevention of Bacteria and Fungi during the Tissue Culture of Dwarf Hygro (Hygrophila polysperma) Explants"

_microorganisms, 2022, doi:10.3390/microorganisms10122476_

Round 1

Reviewer 2 Report

The research might be useful, if authors explain the importance of it. Plant tissue culture technique exists more than 100 years. Is there still such a problem like contamination? Why? Why is it so difficult to solve it? Or is it difficult only for aquatic plants. However according Karatas et al., 2013 Amoklavin is a good way to eradicate contamination.

The research is about contamination in plant tissue culture of Dwarf Hygro, but the most part of introduction is about the history and importance of plant tissue culture, the other part is devoted to biological description of Hygrophila polysperma. It is not clear why did you decide to isolate and identification bacteria during plant tissue culture. Why is it so necessary? Moreover if to take into account a ready protocol of Karatas, I do not understand the reason. Also at the stage of sterilization, it is possible to make many explants (100-200), and if at least 3% will survive and be sterile, this would be a good result.

In addition, it is not clear what did you study. Did you isolate bacteria and fungi just after sterilization or after several rounds of regeneration? In discussion, you speak about endophytes. Did you study endophytes?

Why did not you use antimicrobial compounds Kathon LXE and Vitrofural? Why did you use nanoscale silver, potassium sorbate, sodium benzoate and sodium diacetate?

Round 2

Reviewer 1 Report

After being revised by the authors, the received manuscript basically meets the publishing requirements and is recommended to be published.

Reviewer 2 Report

The research is good. the text is well written with monot mistakes. 

My comments:

Keywords: it would be good to give “contamination” instead of pollution prevention.

Lines 25-33 is better to delete. This is not a review about plant tissue culture.

Line 74 – check the grammar

Line 82 -what kind of leaves? Upper, lower?

Line 86 – what kind of flask did you use for explants? How many explants were put in one flask?

Line 90 – give the manufacturer of the climatic chamber

Line 93 – how many days did it take bacteria to appear?

Line 94 – two bacterial media were used. How did you choose what medium to use for certain bacteria? Why did not you use only one medium? Why did not you use only one medium for all the appearing bacteria? There should be a reason and it is not clear.

Line 101 – what does mean “usually”? Is it less than frequently? if you did not count the % of fungi contaminants, do not use such words. It is better to say “fungi also appeared” And the question is How much time did it take fungi to appear on MS medium?

Line 125 – to or in?

Lines 114 -136 – add a country for a manufacturer

Lines 166 -169 are not necessary. Or give this in introduction.

Lines 169-170 “Currently, rapid propagation of H. polysperma is importantly depends on the tissue culture technique.” What does in mean? Does it depend on type of medium (solid/liquid)? Or does it depend on type of explant for introducing in in vitro?

Figure 1. It would be good to mark bacterial and fungi contamination with arrows of different colors. For example, red for bacteria and blue for fungi.

Line 177 – How rapidly? How many days did it take to cover the medium?

Why does Figure 4 come before Figure 3?

Line 194 – is this figure 1? Also use Figure or Fig. Check this.

Figure 5S – what does red arrow mean?

There should not be a word “respectively” in a caption for figures. The caption for figure 1 is correct and for figure 3 is not.

Lines 313 -314 – in my opinion this is not the best definition of “endophytes”. According to this all the bacteria that just cover the stems are endophytes. I prefer definition that was given by Volk et al 2022 “Minimizing the deleterious effects of endophytes in plant shoot tip cryopreservation”.
